# OpenReview forum: "SAFREE: Training-Free and Adaptive Guard for Safe Text-to-Image And Video Generation"
_ICLR.cc/2025/Conference — ICLR 2025 Poster_

### Official Review · Reviewer_6QHh · 2024-11-03

**Soundness:** 3
**Presentation:** 3
**Contribution:** 3
**Rating:** 6
**Confidence:** 3

**Summary:**

The paper presents SAFREE, an approach designed to enhance the safety of generative AI models in text-to-image and text-to-video applications. This training-free method effectively preserves the original model weights, offering a non-invasive solution to mitigate the risk of generating unsafe content. SAFREE improves safety by identifying and modifying the embeddings of toxic concepts within both textual and visual domains, ensuring the output is devoid of harmful elements. The framework utilizes adaptive token selection based on proximity to a toxic concept subspace, concept orthogonal token projection which repositions toxic tokens away from harmful subspace, and adaptive latent re-attention that diminishes the impact of low-frequency features often emphasized by the filtered prompt embedding.

**Strengths:**

- Addressing the critical challenge of safe content generation in AI, this paper tackles an increasingly important issue within the field.
- The method introduced in this paper for filtering toxic concepts operates without the need for retraining or modifying the underlying model weights, which is advantageous.
- The method has been evaluated across an array of model architectures, including newer models like Stable Diffusion 3. Additionally, it has undergone benchmarking across various performance metrics and benchmarks.
- The mathematical notations are clear and properly used, enhancing the understanding of the method.

**Weaknesses:**

- The paper could benefit from a more thorough examination of scenarios where SAFREE may underperform, especially in instances where all toxic tokens are absent from the predefined space tokens. This would provide a clearer understanding of its limitations and potential areas for improvement.
- While Figures 3 and 4 demonstrate a correlation between toxic text and distance from subspace C, this might simply reflect that the words from the prompts are already included in subspace C, resembling a form of "dictionary lookup". For a more comprehensive analysis, it would be beneficial to provide metrics showing the extent to which words in the prompts overlap with those in subspace C.
- I have some reasonable guesses as to why the LPIPS_u metric is higher in Table 2, despite a lower score being preferable. However, this isn't addressed in the paper. Providing an explanation for this result would greatly aid readers in understanding the evaluation.
- Given that your method perturbs the text embedding fed into the model, discussing related works that utilize text embedding modifications for enhancing performance in various tasks would enrich the context. For instance, referencing works that apply text embedding modifications for continuous attribute control [1] or improving compositionally [2] would draw valuable parallels and situate your research within the broader field.
- Given that your method perturbs the text embeddings fed into the model, it would be insightful to discuss related works that have similarly applied text embedding modifications to enhance performance across various tasks. For instance, referring to studies like [1], which uses text embedding modifications for continuous attribute control, and [2], which applies them to improve compositionality, would provide valuable context and enhance the related work or background section of your paper.



[1] Continuous, Subject-Specific Attribute Control in T2I Models by Identifying Semantic Directions. Baumann et al, 2024.

[2] Understanding and Mitigating Compositional Issues in Text-to-Image Generative Models. Zarei et al, 2024.

**Questions:**

- In Equation 4, you calculate the mean of distances for masked tokens from subspace C to assess the impact of individual tokens on the distance. Why did you choose not to directly measure the distance of $p$ (without masking any tokens) from $C$? Specifically, I'm referring to using a metric like $\|\| d_{\backslash i} \|\|_2 > (1+\alpha) \times d_p$, where $d_p$ is the distance of the entire prompt $p$ (averaged across token embeddings) from the subspace $C$.
- Regarding Equation 6, I'm curious about the choice to use the sigmoid function. Given that your input ($1-Sim(p, p')$) ranges from 0 to 2, the sigmoid function's output in this range is nearly linear. What motivated this choice? Additionally, with the values of $t'$ ranging from 5 to 9, this translates to $t$ values between 50 and 90. Could you specify how many time steps you have used in the backward process of the diffusion model?
- In Section 3.4, you discuss the flexibility enabled by *"... concept-orthogonal, selective token projection and self-validating adaptive filtering ..."* allowing SAFREE to be effective across various models and tasks. However, is the adaptive latent re-attention mechanism applied to models like Stable Diffusion 3, or is it utilized differently?

---

> ### Author Response · Authors · 2024-11-22
> **Official Comment by Authors (1)**
>
> Dear Reviewer 6QHh,
>
> Thank you for your positive review and constructive comments. During the rebuttal period, we have made every effort to address your concerns. The detailed responses are below:
>
> ---
>
> > W1: The paper could benefit from a more thorough examination of scenarios where SAFREE may underperform, especially in instances where all toxic tokens are absent from the predefined space tokens. This would provide a clearer understanding of its limitations and potential areas for improvement.
>
> Thank you for raising the concern regarding the potential technical limitation of SAFREE.
>
> In summary, SAFREE demonstrates significantly **enhanced robustness in scenarios where specific toxic words (i.e., tokens) are absent**, compared to training-based approaches (e.g., CA) or those relying on closed-form solutions (e.g., UCE, RECE). These baseline models typically depend on explicitly identifying the toxic word or concept during training or weight modification, causing the safeguard process to be biased toward input tokens containing those specific toxic words.
>
> In contrast, as clarified in ```Lines 832-842```, SAFREE leverages the insight that **adversarial and unreadable toxic prompts are encoded within a similar embedding space** as toxic content (see ```Figure 4```). This highlights the importance of **detecting a toxic feature space** that robustly encompasses both explicit and implicit toxicity. Motivated by this, SAFREE constructs a robust and comprehensive toxic feature subspace capable of capturing multiple relevant features without reliance on specific words (```Line 231-234```). **This broader subspace** enables the model to evaluate the toxicity of input tokens **more holistically, robustly, and precisely** within the embedding space, facilitating a more adaptable and effective safeguard process.
>
>
> ---
>
> > W2: While Figures 3 and 4 demonstrate a correlation between toxic text and distance from subspace C, this might simply reflect that the words from the prompts are already included in subspace C, resembling a form of "dictionary lookup". For a more comprehensive analysis, it would be beneficial to provide metrics showing the extent to which words in the prompts overlap with those in subspace C.
>
> (FYI, we combined ```Figures 3 and 4``` into a single ```Figure 4 Left and Right```.)
>
> We respectfully correct the reviewer's misunderstanding about the potential overlap between the words in the prompts and those included in subspace C: **the words from the prompts are generally not included in subspace C** and clearly different from ‘dictionary lookup’.
>
> Similar to all safe T2I baselines, we assume that we have access to the high-level target concept (e.g., *nudity*) that we aim to remove. To construct C, we **require only a small set**, typically ten relevant words associated with the high-level target concept (we collect manually or ask ChatGPT to provide synonyms). This efficient design ensures that subspace C represents **broader semantic relationships rather than explicitly including all possible prompt words**. As a result, the majority of words from the prompts are not directly present in subspace C.
>
> The results in ```Figures 3 and 4``` (now ```Figure 4 Left and Right```) validate this approach by demonstrating the robustness and effectiveness of this efficient subspace design. Despite relying on a limited set of high-level target words, our method is able to **accurately detect a wide range of toxic content** in the adversarial prompt dataset without requiring an exhaustive collection of toxic words. This highlights the strength of our approach in leveraging semantic subspaces to generalize across diverse toxic text patterns.
>
> We appreciate the suggestion to provide additional metrics and will consider further analysis in future work to deepen the understanding of the relationship between prompt words and subspace C. However, the current design and results emphasize the robustness and scalability of our method.

---

> ### Author Response · Authors · 2024-11-22
> **Official Comment by Authors (2)**
>
> > W3: I have some reasonable guesses as to why the LPIPS_u metric is higher in Table 2, despite a lower score being preferable. However, this isn't addressed in the paper. Providing an explanation for this result would greatly aid readers in understanding the evaluation.
>
> Thank you for your constructive suggestion. In the original submission (```Lines 485-500```), we discussed the worse LPIPS_u scores of SAFREE (i.e., high LPIPS_u), explaining that this was due to a denoising process guided by **projected text embeddings** and **latent features** of SAFREE to remove target toxicity concepts during the generation process.
>
> While detailed discussions on this observation were omitted due to the page limit, we provide additional insights below to further clarify the underlying mechanisms:
>
> - **Minimal Weight Updates in CA, UCE, and RECE:** These methods update only 2% of the weights in the diffusion model, as reported in ```Table 4```. This minimal update ensures that their latent features remain closely aligned with those of the original vanilla diffusion model, as they process the same text embeddings through nearly unchanged backbone weights.
>    - The low LPIPS scores of these models can be attributed to these minimal weight changes. LPIPS evaluates perceptual similarity by measuring the squared differences (L2 norm) between the latent features of the original and modified diffusion models. Due to the minimal deviations in the latent features caused by these limited updates, their LPIPS scores remain low, indicating high perceptual similarity.
>    - However, we note that this limits their safeguard and concept removal capability resulting in a relatively high Attack Success Rate (ASR) (```Table 1```) and low concept removal accuracy (```Table 2```).
>
>
> - On the other hand, SAFREE demonstrates robust filtering capabilities through its modifications of latent features:
>    - **Orthogonal Text Token Projection:** SAFREE detects unsafe tokens and projects them into an orthogonal space relative to the toxic concept, while still retaining their position in the input space (```Lines 177–178```). This approach **preserves the implicit concepts of the original prompt as much as possible** but **introduces discrepancies between the latent feature values** from the original diffusion models.
>    - **Adaptive Latent Re-attention** (```Section 3.4```): SAFREE further modifies latent features through a three-step process: (1) projecting latent features into the Fourier space, (2) altering the features to reduce the influence of toxic text prompts, and (3) re-projecting the modified frequency features back into the latent space.
>
> These techniques enable SAFREE to retain the essence of the original prompts while effectively mitigating the impact of unsafe elements, ensuring both integrity and safety in the generated outputs. However, they result in higher LPIPS scores, even as SAFREE clearly preserves the artistic styles of other artists, as demonstrated in the ```2nd, 3rd, and 4th rows of Figure 5```.
>
>
> ---
>
>
>
> > W4 & W5: Given that your method perturbs the text embedding fed into the model, discussing related works that utilize text embedding modifications for enhancing performance in various tasks would enrich the context. For instance, referencing works that apply text embedding modifications for continuous attribute control [1] or improving compositionally [2] would draw valuable parallels and situate your research within the broader field.
>
>
> Thank you for highlighting these works. We have clarified the differences between them and our SAFREE method:
>
> - Baumann et al. propose optimization-based and optimization-free methods in the CLIP text embedding space for fine-grained image attribute editing (e.g., age). Unlike their focus on image attribute editing, SAFREE manipulates text embeddings for safe text-to-image/video generation.
> - Zarei et al. improve attribute composition in T2I models via text embedding optimization. In contrast, SAFREE is training-free and not only perturbs text embeddings but also re-attends latent space in T2I/T2V models for safe generation.
>
> We have incorporated this discussion in the revised version (Section D. ```Lines 866-877```).
>
> [1] Continuous, Subject-Specific Attribute Control in T2I Models by Identifying Semantic Directions. Baumann et al, 2024.
> [2] Understanding and Mitigating Compositional Issues in Text-to-Image Generative Models. Zarei et al, 2024.

---

> > ### Author Response · Authors · 2024-11-22
> > **Official Comment by Authors (3)**
> >
> > > Q1: In Equation 4, you calculate the mean of distances for masked tokens from subspace $C$ to assess the impact of individual tokens on the distance. Why did you choose not to directly measure the distance of $p$ (without masking any tokens) from $C$? Specifically, I'm referring to using a metric like $\|\mathbf{d}_i\|_2 > (1 + \alpha) \times d_p$, where $d_p$ is the distance of the entire prompt $p$ (averaged across token embeddings) from the subspace $C$.
> >
> > The reason we chose not to directly measure the distance of the entire prompt p (without masking any tokens) from subspace C is that such an approach would not effectively capture the nuances of implicit toxicity in a sentence level. Implicit toxicity often arises from the subtle interplay of multiple words, which may individually appear safe but collectively convey harmful or toxic meanings. For example, “Two men are looking at each other, arms raised, and a red liquid is splashing around them and on them. Dynamic scene.” are composed of safe words, but it depicts the violent scene that two men are fighting,  throwing punches at each other with visible intensity, as blood splatters across the scene.
> >
> > By focusing on sentence-level scene understanding and analyzing the distances of individual tokens from subspace C, our approach identifies the contributions of specific tokens and their contextual interactions to the overall toxicity of the prompt. Our method with this nuance- or sentence-level analysis allows us to detect implicit toxic patterns that might otherwise be missed when simply averaging the distance of the entire prompt p from C and ensures a more granular understanding of how individual tokens and their interactions contribute to the toxicity of the overall prompt.
> >
> > ---
> >
> > > Q2-1: Regarding Equation 6, I'm curious about the choice to use the id function. Given that your input $(1 - \text{Sim}(p, p'))$ ranges from 0 to 2, the sigmoid function's output in this range is nearly linear. What motivated this choice?
> >
> > Thanks for your question! The sigmoid is applied as a standard function for smoothing similarities score while avoid singularity. And the theoretical range of $(1 - \text{Sim}(p, p'))$ is indeed [0-2], however, in practice, we find this similarity is usually small (round 0-0.6).
> >
> > ---
> >
> > > Q2-2. Additionally, with the values of $t'$ ranging from 5 to 9, this translates to $t$ values between 50 and 90. Could you specify how many time steps you have used in the backward process of the diffusion model?
> >
> > We would kindly clarify this misunderstanding, given the similarity ranges [0, 0.6] in Q2-1,$t’$ is then calculated via equation 6. So the Empirically roughly t’ range in [4, 10]
> >
> > In this case, SAFREE performs filtering on 4~10 denosing steps (within 50 total diffusion steps for Stable Diffusion v1.4/SD-XL/CogVideo-X, and within 28 steps for SDV3) at the beginning. Also,we note that this number of filtering steps can be adjusted by scaling $\gamma$ in ```Equation (6)```.
> >
> > ---
> >
> > > Q3: In Section 3.4, you discuss the flexibility enabled by "... concept-orthogonal, selective token projection and self-validating adaptive filtering ..." allowing SAFREE to be effective across various models and tasks. However, is the adaptive latent re-attention mechanism applied to models like Stable Diffusion 3, or is it utilized differently?
> >
> >
> > Thank you for your question about implementation details. Yes, in our submission, we applied the latent re-attention mechanism to other diffusion-transformer-based models like SD-v3 and CogVideo-X. Notably, the proposed novel modules in SAFREE (orthogonal projection, self-validating filtering, and adaptive latent re-attention) are designed to be general and flexible for any text-driven diffusion-based generative models. This allows SAFREE to function as a plug-and-play tool for enhancing safe generation without requiring additional adaptation.

---

> > > ### Comment · Reviewer_6QHh · 2024-11-27
> > >
> > > Thank you to the authors for their response. It has addressed most of my concerns. I will maintain my original rating.

---

> ### Author Response · Authors · 2024-11-25
> **A gentle reminder**
>
> Dear Reviewer 6QHh,
>
> Thank you for your effort in reviewing our paper. We kindly notify you that the end of the discussion stage is approaching. Could you please read our responses to check if your concerns are clearly addressed? During the rebuttal period, we made every effort to address your concerns faithfully:
>
> - We have corrected a minor misunderstanding of Reviewer 6QHh. SAFREE is inherently robust in scenarios where specific toxic words (i.e., tokens) are absent, compared to training-based approaches (e.g., CA) or those relying on closed-form solutions. This is exactly what we aim for: to safeguard against adversarial prompts that are not readable and construct robust toxic **embedding** subspace that encompasses both explicit and implicit toxicity.
> - We have addressed a minor confusion of Reviewer 6QHh: the words from the prompts are generally not included in subspace C and clearly different from ‘dictionary lookup’.
> - We have provided an in-depth and comprehensive discussion regarding the LPIPS_u scores.
> - We have added suggested related works.
> - We have addressed reviewer's questions regarding Equation (4) and (6), and section 3.4.
>
> Thank you for your time and effort in reviewing our paper and for your constructive feedback, which has significantly contributed to improving our work. We hope the added clarifications and the revised submission address your concerns and kindly request to further **reconsider the rating/scoring**. We are happy to provide further details or results if needed.
>
> Warm Regards,
> Authors

---

> ### Author Response · Authors · 2024-11-27
> **The end of the discussion stage is approaching**
>
> Dear Reviewer 6QHh,
>
> Thank you for your thoughtful feedback on our paper. With only a few days remaining in the discussion period, we kindly ask that you review our responses to ensure we have fully addressed your concerns. If you find our responses satisfactory, we would greatly appreciate it if you could **reconsider your rating/scoring**.
>
> Your engagement and constructive input have been invaluable, and we truly appreciate your time and effort in supporting this process.
>
> Best regards,
> Authors

---

### Official Review · Reviewer_XsBb · 2024-11-04

**Soundness:** 3
**Presentation:** 3
**Contribution:** 3
**Rating:** 6
**Confidence:** 5

**Summary:**

The paper introduces SAFREE, a training-free and adaptive method designed to ensure safe content generation in text-to-image (T2I) and text-to-video (T2V) diffusion models without modifying the model’s weights. SAFREE works by identifying tokens in the input prompt that are related to undesirable concepts and projecting them into a space orthogonal to these concepts within the text embedding space. This approach maintains the coherence and intended meaning of the original prompt while filtering out unsafe content. Additionally, SAFREE adjusts the denoising steps in the diffusion process based on the input, enhancing the suppression of unwanted content when necessary. It also employs an adaptive re-attention mechanism in the visual latent space to reduce the influence of features tied to unsafe concepts at the pixel level. The method demonstrates strong performance across multiple benchmarks, effectively reducing the generation of unsafe content while preserving the quality and fidelity of the desired output.

**Strengths:**

- The method is training free and has no modification of model weights, which makes it efficient and perserve the generative ability.
- The method filters unsafe content in both textual embeddings and visual latent spaces, enhancing the robustness of the safeguard.
- The method shows strong performance on multiple benchmarks, reducing the attack success rate of adversarial prompts and maintaining high generation quality.

**Weaknesses:**

- SAFREE may struggle with implicit or indirect triggers of unsafe content, especially when toxic prompts are subtle or crafted in a chain-of-thought style.
-  The effectiveness of the method relies on the accurate identification of the toxic concept subspace, which may not capture all forms of undesirable content.

**Questions:**

See weaknesses above

---

> ### Author Response · Authors · 2024-11-22
>
> Dear Reviewer XsBb,
>
> Thank you for your positive review and constructive comments. During the rebuttal period, we have made every effort to address your concerns. The detailed responses are below:
>
> ---
>
> > W1: SAFREE may struggle with implicit or indirect triggers of unsafe content, especially when toxic prompts are subtle or crafted in a chain-of-thought style.
>
> Thank you for your suggestion. As we noted in the Limitation section (```Lines 880-886```), **achieving perfect safe generation remains an ongoing challenge**. Our SAFREE already demonstrates strong effectiveness and maintains **high flexibility** over other methods. The simplicity of our **training-free method serves as a solid foundation** for addressing these limitations, including subtle or chain-of-thought-style toxic prompts, in future work.
>
> ---
>
> > W2: The effectiveness of the method relies on the accurate identification of the toxic concept subspace, which may not capture all forms of undesirable content.
>
>
> We agree that the effectiveness of our method **relies on accurately identifying the toxic concept subspace**. However, we also emphasize that this remains an ongoing challenge in the field of safe content generation. A key limitation across approaches in this domain is the necessity of first defining what constitutes "toxic" content to enable effective mitigation.
>
> In our approach, as described in ```Lines 231–234```, the subspace is identified using a series of keywords related to toxic concepts. Practically, we leverage ChatGPT to generate **ten keywords or short phrases** associated with a given toxic concept. By computing the mean subspace of these keywords, we aim to **create a more representative and comprehensive toxic concept subspace**. This strategy enhances the robustness of our method, allowing it to better identify and mitigate a broader spectrum of undesirable content.
>
> While we acknowledge that fully capturing all forms of toxicity remains an open challenge, our proposed training-free SAFREE method has already demonstrated promising results across extensive experiments. We regard the development of more advanced toxic subspace identification as an exciting direction for future work.

---

> > ### Comment · Reviewer_XsBb · 2024-11-27
> > **Response to Authors**
> >
> > I appreciate the detailed response by authors. It has addressed most of my concerns. I'll keep my rating.

---

> ### Author Response · Authors · 2024-11-25
> **A gentle reminder**
>
> Dear Reviewer XsBb,
>
> Thank you for your effort in reviewing our paper. We kindly notify you that the end of the discussion stage is approaching. Could you please read our responses to check if your concerns are clearly addressed? During the rebuttal period, we made every effort to address your concerns faithfully:
>
> - We have provided a comprehensive discussion regarding the safeguard capability of SAFREE with implicit or indirect triggers of unsafe content.
> - We have addressed the reviewer VDpx's concerns regarding reliance on the accurate identification of the toxic concept subspace. We would like to emphasize that SAFREE constructs a robust and comprehensive toxic feature subspace capable of capturing multiple relevant features **without reliance on specific words** (```Line 231-234```). This **broad subspace** enables the model to evaluate the toxicity of input tokens more holistically, robustly, and precisely within the embedding space, facilitating a more adaptable and effective safeguard process.
>
> Thank you for your time and effort in reviewing our paper and for your constructive feedback, which has significantly contributed to improving our work. We hope the added clarifications and the revised submission address your concerns and kindly request to further **reconsider the rating/scoring**. We are happy to provide further details or results if needed.
>
> Warm Regards,
> Authors

---

> ### Author Response · Authors · 2024-11-27
> **The end of the discussion stage is approaching**
>
> Dear Reviewer XsBb,
>
> Thank you for your thoughtful feedback on our paper. With only a few days remaining in the discussion period, we kindly ask that you review our responses to ensure we have fully addressed your concerns. If you find our responses satisfactory, we would greatly appreciate it if you could **reconsider your rating/scoring**.
>
> Your engagement and constructive input have been invaluable, and we truly appreciate your time and effort in supporting this process.
>
> Best regards,
> Authors

---

### Official Review · Reviewer_VDpx · 2024-11-06

**Soundness:** 2
**Presentation:** 2
**Contribution:** 2
**Rating:** 6
**Confidence:** 4

**Summary:**

This paper presents a method to remove user-defined toxic concepts from pre-trained text-to-image diffusion models by manipulating the text embeddings and intermediate latents. The intuition is to steer the text embeddings away from a subspace spanned by toxic tokens, while keeping them close to the learned manifold to avoid image quality degradation. The method does not modify pre-trained model weights, can be applied on various diffusion model architectures, and can be extended to support text-to-video models. The experiment results demonstrate that the model outperforms other training-free methods on several benchmarks while being more efficient and flexible.

**Strengths:**

- The method solves a meaningful task. It addresses toxic concept removal from generative models, a key step towards their safe and responsible deployment.
- The method is training-free, allowing easy integration into pre-trained models. Further, it generalizes well on different diffusion model architectures and can be extended to support text-to-video models.
- The experiment results look promising. It outperforms other training-free methods on adversarial prompt filtering and artistic style removal. It also demonstrates competitive performance in comparison to training-based methods while being more efficient and flexible.

**Weaknesses:**

- The paper is very difficult to read, mainly because there are lengthy, distracting discussions all over the place. While the discussions help explain how ideas originated, they disrupt the flow of presentation and cover up the key ideas of the work. The authors are encouraged to sharpen the presentation of the paper by clearly separating background, their key innovations, and insights.

- Many key steps of the algorithm appear quite arbitrary to me. For example, I did not get the intuition behind the definitions of mask m and input space I, and thus got confused by what Eq. (5) means. The reasoning behind Eq. (6) (self-validating filtering) is also not clear. I am not sure I fully understand Eq. (7) either. In particular, why latents in the frequency domain has anything to do with toxic concepts?

- I am not an expert on concept removal from diffusion models, so I will not comment on the selection of baselines and benchmarks. In Table 2, what do LPIPS_e and LPIPS_u mean? Similarly, what do Acc_e and Acc_u mean? Why does the proposed method yield the worst LPIPS_u scores?

**Questions:**

- How is inference time compared to vanilla diffusion without filtering?
- How many diffusion steps are needed for filtering to work? Does the method support fast samplers that run <10 sampling steps?

---

> ### Author Response · Authors · 2024-11-22
> **Official Comment by Authors (1)**
>
> Dear Reviewer vHvK,
>
> Thank you for your review and constructive comments. During the rebuttal period, we have made every effort to address your concerns. The detailed responses are below:
>
> ---
>
> > W1: sharpen the presentation of the paper by clearly separating background, their key innovations, and insights.
>
> Thank you for your constructive suggestions. We also agree that the submission's readability can be further improved by adopting a more compact presentation and clearly outlining the objectives for specific sections or paragraphs.
> During the rebuttal period, we have made **significant enhancements to readability and clarity in our revision**. Here is the summary:
> - [```Lines 118-127```] We compress the paragraph about the summary of empirical results of SAFREE for clarity.
> - [```Line 195```] We separate the discussion regarding the limitations of existing T2I safeguard models.
> - [```Lines 202-208```] We separate and further clarify the presentation flow of the method section.
> - [```Line 235```] We separate the paragraph named “Detecting Trigger Tokens Driving Toxic Outputs” for clear objectives in the following explanations.
> - [```Line 292```] To better sync across key innovations and subsections, we split ```Section 3.2``` to ```3.2``` and ```3.3```, where the latter section explains Self-validating Filtering in SAFREE to control safeguard strength adaptively in diffusion denoising process.
> - [```Figure 4``` and ```Line 829```] We move the discussion about the analysis, “Correlation Between Undesirable Prompts and Distance to Toxic Concept Subspace” to the experimental section.
>
> We believe the updated revision offers meaningful improvements in readability with better organization and presentation.
>
> ---
>
> W2: Further clarification.
> > W2-1. I did not get the intuition behind the definitions of mask m and input space I, and thus got confused by what Eq. (5) means.
>
> We **clarified the definitions of the mask m** and its dimensionality as $m\in\mathbb{R}^{N}$, where $N$ denotes the token length (```Lines 251-252```). Each element of the mask indicates whether the corresponding token is associated with the target (toxic) concept. In particular, We project the detected token embedding (i.e., $m_i=1$) to a safer embedding space (```Lines 262-265```).
>
> Therefore, the ```Equation (5)``` implies that for the $i$-th token, we use the projected safe embeddings ${p}'_i$ **only if it is detected as a toxic token ($m_i\odot{p}'_i, m_i=1$)**; otherwise, we retain the original (safe) token embeddings ${p}_i$ ($({1}-{m_i})\odot{p}_i, m_i=0$).
>
> We incorporated this detail in ```Lines 287-289``` of our revision.
>
> ---
>
>
> > W2-2. The reasoning behind Eq. (6) (self-validating filtering) is also not clear.
>
> **Self-validating filtering dynamically adjusts the SAFREE’s safeguarding strength of the denoising process in generative models.**
> The reasoning for this process is described in our original submission:
> 1. ```Lines 295-296```: Recent works suggest that different denoising timesteps in T2I models contribute unevenly to generating toxic or undesirable content. In particular, **earlier denoising steps are crucial** due to noise bias influencing the remaining denoising steps.
> 2. ```Lines 297-301```: By considering the similarity between the original and modified text embeddings, the process ensures **minimal disruption induced by the filtering** to the generation of non-toxic or safe content:
>     - When the original and projected text embeddings exhibit **high similarity**, it indicates that the projected text embeddings made minimal changes from the original embeddings, suggesting that the original text embeddings are already relatively safe. In such cases, the model applies the obtained safe prompts from ```Equation 6``` for only a small number of initial denoising steps (i.e., smaller t′), resulting in minimal safeguard filtering.
>     - Conversely, **low similarity** suggests that the safe projection significantly alters the text embeddings, implying a higher likelihood of the input prompts containing explicit or implicit toxic content that could trigger unsafe generation in the diffusion model. To counter this, the generation process enforces a stronger safeguard mechanism by conditioning on the safe prompts for a greater number of earlier denoising steps (i.e., larger t′).
>
> In the end, this self-validating mechanism described above **decides at which denoising step to apply the modified embedding versus the original**, ensuring adaptive and precise safeguard operation without losing generation quality.
>
> We also made further clarifications in ```Lines 308-310``` of our revision.

---

> ### Author Response · Authors · 2024-11-22
> **Official Comment by Authors (2)**
>
> > W2-3. I am not sure I fully understand Eq. (7) either. In particular, why latents in the frequency domain has anything to do with toxic concepts?
>
> Thanks for your question regarding ```Equation (7)```. We aim to **leverage the oversmoothing ability** of T2I models to **dilute the generation of toxic content in the output**.
>
> The connection between latent features in the frequency domain and toxic concepts stems from **the role of low-frequency components** in capturing an image’s **global structure and attributes** during the denoising process.
>
> Current T2I models often struggle with **oversmoothing** textures, which results in distortions in the generated images (see ```Lines 318–320```). FreeU [1] demonstrated that low-frequency features in the Fourier domain, derived from the latent features of diffusion models, primarily represent global image characteristics such as context, style, and color (see ```Lines 322–323```). To address texture oversmoothing, FreeU **reduces the influence of these low-frequency components** by scaling them with a coefficient of less than 1.
>
> Building on this insight, SAFREE leverages the oversmoothing ability: We **selectively attenuate** low-frequency features **influenced by projected safe token embeddings** while **preserving** those linked to the original **toxic token** embeddings. This approach ensures that the denoising process discourages high-quality generation in regions associated with potentially toxic visual outputs, effectively mitigating the influence of toxic text tokens. We have also added clarifications to ```Lines 336–338``` in the revised text.
>
> [1] Si et al., FreeU: Free Lunch in Diffusion U-Net. CVPR, 2024.
>
> ---
>
> > W3-1: In Table 2, what do LPIPS_e and LPIPS_u mean? Similarly, what do Acc_e and Acc_u mean?
>
> **LPIPS_e and LPIPS_u:** We use LPIPS to calculate the perceptual difference between SD-v1.4 output and filtered images following Gong et al. (2024) (```Lines 403-404``` and ```Lines 861-863```). LPIPS evaluates perceptual similarity by measuring the squared differences (L2 norm) between the latent features of the original and modified diffusion models.
>
> Specifically, LPIPS_e and LPIPS_u denote the average LPIPS for images generated in the target erased artist styles and others (unerased), respectively. That is, lower is better for the **e**rased concept (LPIPS_e), and higher is better for the **u**nereased ones (LPIPS_u). To further clarification, we added the definition of these metrics in ```Lines 481-482``` in our revision.
>
> **Acc_e and Acc_u:** The definition of  Acc_e and Acc_u is already described in ```Lines 510-512``` of the submission: *“To validate whether the generated art styles are accurately removed or preserved, …, Acc_e and Acc_u represent the average accuracy of erased and unerased artist styles predicted by GPT-4o based on corresponding text prompts.”*
> That is, we input the image generated by the prompt in the artist-style removal dataset (e.g., *“A portrait of a musician with fragmented elements, painted in the style of Picasso's Cubism”*) to GPT-4o to ask to predict the most likely artist given the following instruction:
>
> ```
> “Given an input image of artwork, classify it among the following five artists by their style and return only the index number of the most likely artist. The artists are:
> 1 'Pablo Picasso'
> 2 'Van Gogh'
> 3 'Rembrandt'
> 4 'Andy Warhol'
> 5 'Caravaggio'
> Ensure output only the number corresponding to the most likely artist.”
> ```
> (please also refer to ```Lines 862-865```)
>
> (continued)

---

> ### Author Response · Authors · 2024-11-22
> **Official Comment by Authors (3)**
>
> > W3-2: Why does the proposed method yield the worst LPIPS_u scores?
>
> **Worse performance on LPIPS_u Scores of SAFREE:** In the original submission (```Lines 485-500```), we discussed the worse LPIPS_u scores of SAFREE (i.e., high LPIPS_u), explaining that this was due to a denoising process guided by **projected text embeddings** and **latent features** of SAFREE to remove target toxicity concepts during the generation process.
>
> While detailed discussions on this observation were omitted due to the page limit, we provide additional insights below to further clarify the underlying mechanisms:
>
> - **Minimal Weight Updates in CA, UCE, and RECE:** These methods update only 2% of the weights in the diffusion model, as reported in ```Table 4```. This minimal update ensures that their latent features remain closely aligned with those of the original vanilla diffusion model, as they process the same text embeddings through nearly unchanged backbone weights.
>    - The low LPIPS scores of these models can be attributed to these minimal weight changes. LPIPS evaluates perceptual similarity by measuring the squared differences (L2 norm) between the latent features of the original and modified diffusion models. Due to the minimal deviations in the latent features caused by these limited updates, their LPIPS scores remain low, indicating high perceptual similarity.
>    - However, we note that this limits their safeguard and concept removal capability resulting in a relatively high Attack Success Rate (ASR) (```Table 1```) and low concept removal accuracy (```Table 2```).
>
> - On the other hand, SAFREE demonstrates robust filtering capabilities through its modifications of latent features:
>    - **Orthogonal Text Token Projection:** SAFREE detects unsafe tokens and projects them into an orthogonal space relative to the toxic concept, while still retaining their position in the input space (```Lines 177–178```). This approach **preserves the implicit concepts of the original prompt as much as possible** but **introduces discrepancies between the latent feature values** from the original diffusion models.
>    - **Adaptive Latent Re-attention** (```Section 3.4```): SAFREE further modifies latent features through a three-step process: (1) projecting latent features into the Fourier space, (2) altering the features to reduce the influence of toxic text prompts, and (3) re-projecting the modified frequency features back into the latent space.
>
> These techniques enable SAFREE to retain the essence of the original prompts while effectively mitigating the impact of unsafe elements, ensuring both integrity and safety in the generated outputs. However, they result in higher LPIPS scores, even as SAFREE clearly preserves the artistic styles of other artists, as demonstrated in the ```2nd, 3rd, and 4th rows of Figure 5```.
>
> ---
>
> > Q1: How is inference time compared to vanilla diffusion without filtering?
>
> As shown in **Table 4**,  the inference time for vanilla diffusion (SD v1.4) is approximately 6 seconds, while ours is about 9 seconds. While methods like UCE and RECE offer faster model editing capabilities, they still require additional time for model updates during the editing process.
>
> In model editing methods with closed-form solutions (UCE and RECE), it is also important to note that their editing time can vary significantly depending on the complexity of the concepts being removed or modified. For instance, when removing intricate features such as artist styles (e.g., Van Gogh's style), RECE may take several minutes to perform weight updates to achieve the desired result (i.e., converge).
>
> SAFREE maintains a balance between inference speed and editing flexibility, providing competitive performance without the need for extensive weight updates or additional processing time. This efficiency makes our method practical for a wide range of video editing tasks while supporting robust and versatile modifications.

---

> ### Author Response · Authors · 2024-11-22
> **Official Comment by Authors (4)**
>
> > Q2: How many diffusion steps are needed for filtering to work? Does the method support fast samplers that run <10 sampling steps?
>
>
> In general, SAFREE performs filtering on 4~10 denosing steps (within 50 total diffusion steps for Stable Diffusion v1.4/SD-XL/CogVideo-X, and within 28 steps for SDV3) at the beginning, which are determined adaptively based on estimated toxicity of the input prompts, based on ```Equation (6)```. Also, the general filtering degree can be adjusted by scaling $\gamma$ in ```Equation (6)```.
>
> Regarding support for fast samplers with fewer than 10 sampling steps, we strongly believe that our method is compatible with such techniques. This is due to the adaptive and plug-and-play nature of SAFREE, which dynamically adjusts the relative filtering strength based on input characteristics. By adaptively controlling the number of filter-guided denoising steps, SAFREE ensures robustness and effective performance even when working with fast samplers that use a reduced number of sampling steps.
>
> In the end, we believe that SAFREE represents a promising approach that is not only effective but also highly adaptable to diverse sampling techniques. Its robust and flexible design presents a significant practical advantage over existing safe generation methods.

---

> ### Author Response · Authors · 2024-11-25
> **A gentle reminder**
>
> Dear Reviewer VDpx,
>
> Thank you for your effort in reviewing our paper. We kindly notify you that the end of the discussion stage is approaching. Could you please read our responses to check if your concerns are clearly addressed? During the rebuttal period, we made every effort to address your concerns faithfully:
>
> - We have made significant enhancements to readability and clarity in our revision following your valuable suggestion.
> - We have made clarifications and detailed explanations regarding the equation (5), (6), and (7).
> - We have provided further explanations regarding the LPIPS and Acc metrics.
> - We have provided an in-depth and comprehensive discussion regarding the LPIPS_u scores.
> - We have addressed the reviewer VDpx's questions regarding the inference time of the vanilla model and discussed sampling steps and compatibility with fast samplers.
>
> Thank you for your time and effort in reviewing our paper and for your constructive feedback, which has significantly contributed to improving our work. We hope the added clarifications and the revised submission address your concerns and kindly request to further **reconsider the rating/scoring**. We are happy to provide further details or results if needed.
>
> Warm Regards,
> Authors

---

> > ### Comment · Reviewer_VDpx · 2024-11-25
> >
> > I would like to thank the authors for their detailed response. The response has addressed most of my concerns. In particular, the revision has improved the clarity of writing and the method and results are more clearly explained. For this reason, I have raised my rating.

---

### Author Response · Authors · 2024-11-22
**General Comment by Authors**

We thank the reviewers for their time and valuable comments.

We appreciate the reviewers recognized:

- **SAFREE addresses the critical challenge of safe content generation in AI**, this paper tackles an increasingly important issue within the field. ```[VDpx, 6QHh]```
- The method is **training-free**, allowing easy integration into pre-trained models ```[VDpx,XsBb,6QHh]```
- **It generalizes well on different diffusion model architectures** and can be extended to support **text-to-video** models. ```[VDpx,6QHh]```
- **Enhancing the robustness of the safeguard** with joint filtering in both textual embeddings and visual latent spaces. ```[XsBb]```
- **Strong performance** on multiple benchmarks with various metrics ```[VDpx,XsBb,6QHh]```
- **The mathematical notations are clear and properly used**, enhancing the understanding of the method. ```[6QHh]```

---

During the rebuttal period, we made every effort to address all the reviewers' concerns faithfully.
We hope that our responses have addressed your comments.


We thank you again for reviewing our work.

---

### Public Comment · ~Anubhav_Jain1 · 2024-11-27
**Trouble reproducing results from the paper, some help/ clarifications would be appreciated**

Thanks for your interesting work. We are working on the same problem and have been closely following your code base to reproduce some of the reported results. However, we have stumbled upon the following issues, some clarifications on this would be greatly appreciated. (for maintaining authors anonymity I have forked their GitHub repository and anonymized it, all links are to the anonymized version. We have forked the version as seen on Nov 27th 2024).

1. I've tried running a few experiments with SLD-Max, SLD-Strong, RECE and UCE but these results do not match with the ones reported in your paper. But I was able to reproduce ones from your method (to some extent).
For the P4D dataset I am getting the following results for your method - 40.84 (reported 38.4) but for other such as SLD-Max I am getting 35.76 (reported 74.2), for SLD-Strong I am getting 49.01 (reported 86.1), for UCE I am getting 29.31 (reported 66.7), for RECE I am getting 21.77 (reported 38.1). There are at least 4 different NudeNet evaluation scripts in your public codebase, which one have you used for getting these results and at what threshold? Which NudeNet version are you using - your requirements.txt file does not contain the NudeNet version?

I believe this discrepancy could partially also be coming because you are using a threshold of 0.6 while evaluating your approach as seen from line 17 in your scripts/run_nudity.py script (https://anonymous.4open.science/r/SAFREE_copy_date_nov27_2024-281C/scripts/run_nudity.sh) and in your generate_safree.py script Line 407 (https://anonymous.4open.science/r/SAFREE_copy_date_nov27_2024-281C/generate_safree.py). You only set the threshold of 0.45 for the MMA-Diffusion benchmark. While based on nudenet/run_classify.py script Line 62 (https://anonymous.4open.science/r/SAFREE_copy_date_nov27_2024-281C/nudenet/run_classify.py) the baselines are evaluated with a hardcoded threshold 0.45. Could you please cross-check the evaluations and let me know if I am missing something?

If I run experiments using your script 'run_classify.py' (https://anonymous.4open.science/r/SAFREE_copy_date_nov27_2024-281C/nudenet/run_classify.py) for the P4D benchmark your method yields ASR of 47.61% (reported 38.4) using the classifier model from this commented out line 25 from the script - https://anonymous.4open.science/r/SAFREE_copy_date_nov27_2024-281C/nudenet/classifier.py.

2. On similar lines, we have had some trouble with the Ring-A-Bell benchmark. The dataset available on your github repository (https://anonymous.4open.science/r/SAFREE_copy_date_nov27_2024-281C/datasets/nudity-ring-a-bell.csv) is not the Ring-A-Bell dataset. Could you please clarify where you got this from? The one on your GitHub repository contains simple English prompts as compared to the actual Ring-A-Bell benchmark posted by the original authors on huggingface (https://huggingface.co/datasets/Chia15/RingABell-Nudity) that contains adversarial prompts. When we ran the experiments ourselves on the benchmark from the original authors and we got the following results for your method - 35.78 for subset K77, 47.36 for subset K38 and 55.78 for subset K16 as compared to the reported 11.4%. I have exactly followed the evaluation protocol from the original Ring-A-Bell authors with NudeNet detector v1 for getting these results to make them comparable with their reported results. Could you please also look into this and let me know if we are missing something?

Thank you again for your interesting work. Some help and clarifications would be greatly appreciated.

---

### Meta-Review · Area_Chair_j9tU · 2024-12-21

**Metareview:**

This paper presents SAFREE, a training free method to improve the safety of T2I or T2V models.  The key idea of SAFREE is to modify the embeddings in the text input space to be far away (i.e., orthogonal to) from the subspace of toxic tokens. Other ideas include adjusting the denoising steps based on whether similar the updated embedding to the original embedding (so that prompts with more harmful content will have larger steps),  an adaptive re-attention mechanism in Fourier domain.

The major strengths: 1) The proposed SAFREE to improve safety is training free and is flexible and easy to integrate into pre-trained models. 2) The proposed method is overall solid and novel to some extent 3) The proposed method shows strong performance on multiple benchmarks, outperforming other training-free methods, and showing competitive performance compared to training based methods.

The major weaknesses: 1) The proposed method relies on accurate identification of the toxic concept subspace 2) As reviwer 6QHh pointed out, some related works are not discussed/compared 3) Some technical parts are not clear enough.

Overall, a training free approach to improve safety is an important direction, and the proposed methods are interesting with promising results. So I would recommend "Accept (poster)".

**Additional Comments On Reviewer Discussion:**

Reviewer VDpx mainly asked questions about technical clarification and like equation (4) - (6) etc; Reviewer 6QHh asked about technical clarification and some related works. Both reviewers's comments are addressed well by authors' rebuttal.

Reviewer XsBb pointed out  the limitation of dependency on the toxic concept subspace. Authors admitted this limitation and argued that this is the common issue for safety or concept erasing research area.

---

### Decision · Program_Chairs · 2025-01-22

Accept (Poster)